# MoRE: A Mixture of Low-Rank Experts for Adaptive Multi-Task Learning

## Abstract

With the rapid development of Large Language Models (LLMs), Parameter-Efficient Fine-Tuning (PEFT) methods have gained significant attention, which aims to achieve efficient fine-tuning of LLMs with fewer parameters. As a representative PEFT method, Low-Rank Adaptation (LoRA) introduces low-rank matrices to approximate the incremental tuning parameters and achieves impressive performance over multiple scenarios. After that, plenty of improvements have been proposed for further improvement. However, these methods either focus on single-task scenarios or separately train multiple LoRA modules for multi-task scenarios, limiting the efficiency and effectiveness of LoRA in multi-task scenarios. To better adapt to multi-task fine-tuning, in this paper, we propose a novel **M**ixture **o**f Low-**R**ank **E**xperts (MoRE) for multi-task PEFT. Specifically, instead of using an individual LoRA for each task, we align different ranks of LoRA module with different tasks, which we named *low-rank experts*. Moreover, we design a novel adaptive rank selector to select the appropriate expert for each task. By jointly training low-rank experts, MoRE can enhance the adaptability and efficiency of LoRA in multi-task scenarios. Finally, we conduct extensive experiments over multiple multi-task benchmarks along with different LLMs to verify model performance. Experimental results demonstrate that compared to traditional LoRA and its variants, MoRE significantly improves the performance of LLMs in multi-task scenarios and incurs no additional inference cost. We also release the model and code to facilitate the community[1].

## 1 Introduction

Recent advancements in Large Language Models (LLMs) have revolutionized various domains, offering unprecedented performance across numerous tasks (Devlin et al., 2019; Raffel et al., 2020; Brown et al., 2020; Touvron et al., 2023). Plenty of tuning strategies are designed to extend the application of LLMs, such as Instruction Tuning (Wei et al., 2022; Zhang et al., 2023b), Continual Pre-Training (Ke et al., 2023), and Parameter-Efficient Fine-Tuning (PEFT) (Houlsby et al., 2019; Liu et al., 2023b; 2022; Lester et al., 2021; Li & Liang, 2021; Hu et al., 2022). Among these strategies, PEFT has drawn the most attention due to its fewer parameter tuning and lower computational cost. As the representative PEFT method, Low-Rank Adaptation (LoRA) (Hu et al., 2022) introduces low-rank matrices to approximate the incremental tuning parameters and demonstrate good performance in many scenarios, which has become a standard paradigm for LLM fine-tuning and inspired many improvements (Liu et al., 2024; Valipour et al., 2023; Ding et al., 2023).

Table 1: LoRA-based Fine-tuning Performance of T5-base with varying ranks on different tasks

| Task/Rank | r=1 | r=2 | r=4 | r=8 | r=16 | r=32 |
|---|---|---|---|---|---|---|
| MRPC | **89.7** | 89.2 | 88.7 | 89.2 | 89.2 | 89.5 |
| RTE | 77.6 | 78.7 | **80.5** | 77.6 | 80.1 | 79.1 |
| SST-2 | 94.4 | 94.6 | **94.8** | 94.5 | 94.4 | 94.5 |
| CoLA | 60.9 | 60.0 | 61.9 | **63.3** | 62.3 | 60.5 |

---

[1]https://anonymous.4open.science/r/MoRE-3C37

Despite the achieved progress, LoRA relies on a fixed and unalterable intrinsic rank, making it not flexible enough in multi-task scenarios. Taking Table 1 as an example, when dealing with different tasks, LoRA requires different ranks to achieve the best performance (e.g., best ranks for MRPC and CoLA tasks are 1 and 8). Considering the high computational cost and storage cost of LLM fine-tuning, training multiple LoRA modules is sub-optimal for applying LLMs to multi-task scenarios. Meanwhile, searching the best rank of LoRA during LLM fine-tuning is also time-consuming and computationally expensive (Valipour et al., 2023), which highlights the limitations of a one-size-fits-all approach in LoRA. This phenomenon also emphasizes the need for adaptive mechanisms that dynamically adjust ranks based on task requirements.

To overcome the limitations of fixed ranks in LoRA, one promising direction is to explore adaptive mechanisms. For example, DyLoRA (Valipour et al., 2023) dynamically trained all ranks during training to avoid separate rank tuning for each task. AdaLoRA (Zhang et al., 2023a) allocated the parameter budget based on the importance scores of the weight matrices and pruned insignificant singular values to exclude unimportant rank spaces. SoRA (Ding et al., 2023) introduced a trainable gating unit and used proximal gradient descent to optimize the sparsity of the update matrices, thereby dynamically adjusting the intrinsic rank size during training. While these improvements enable dynamic adjustment of rank space, they are primarily designed for single-task scenarios. They do not consider the distinctions and connections among different tasks in multi-task scenarios, prohibiting the effectiveness of LoRA in multi-task scenarios.

In the meantime, there also exist other strategies that try to exploit the connections among different tasks. However, they are still far from satisfied. For example, HyperFormer (Mahabadi et al., 2021) enhanced adapter-based methods by utilizing a shared hypernetwork to facilitate cross-task knowledge sharing, while incorporating task-specific adapters to tailor the model for individual tasks. However, they face limitations due to their inherent performance constraints and additional inference latency. Prompt Tuning methods (Vu et al., 2022; Asai et al., 2022; Wang et al., 2023b) are proposed to use learned prompts on source tasks to initialize the learning of target tasks. Despite the effectiveness, these approaches typically require a two-stage training process (i.e., first on the source task and then on the target task), which requires higher data quality and results in training efficiency decrease. Meanwhile, parallel LoRA strategies (Wang et al., 2023a; Li et al., 2024; Liu et al., 2023a; Huang et al., 2023) can effectively address the above shortcoming, offering a better adaptability in multi-task scenarios. Nonetheless, the usage of parallel LoRA modules increases the overall parameter count and resource consumption, contradicting the original purpose of LoRA to reduce the training parameters. Thus, one important question should be considered: "**How to achieve efficient LLM fine-tuning in multi-task scenarios remains challenging**".

To this end, in this paper, we design a novel Mixture of Low-Rank Experts (MoRE) for efficient LLM fine-tuning in multi-task scenarios. Since different tasks require different ranks of LoRA, we propose to build connections between the ranks and the tasks in a Mixture-of-Expert (MoE) manner. Specifically, we propose to treat each rank in the LoRA module as an expert and design a novel *Adaptive Rank Selector*. Thus, *the different experts corresponding to different tasks can share common information and maintain distinctive information simultaneously* (i.e., the ranks $r_i$ and $r_j$ can share some common parameters). Meanwhile, our proposed selector uses a gating mechanism to select the appropriate rank expert for each task. Moreover, to fully exploit the distinctions and connections among different tasks for accurate rank selection, we develop a novel *CL-based Task Embedding* module, which assigns a task embedding to each task and uses a Contrastive Learning (CL) optimization to ensure the quality of learned task embeddings. Furthermore, we incorporate the *Balanced Dataset Sampling strategy* to address the severe dataset imbalance in multi-task scenarios. Along this line, MoRE can fully exploit the potential of LoRA and realize efficient LLM fine-tuning in multi-task scenarios. Finally, we conduct extensive experiments on multi-task benchmarks to validate the effectiveness of MoRE. Experimental results demonstrate the efficiency of MoRE in multi-task and low-resource transfer scenarios.

## 2 RELATED WORK

### 2.1 PARAMETER-EFFICIENT FINE-TUNING (PEFT)

PEFT methods are designed to adapt LLMs to new tasks with minimal additional parameters. Representative works include BitFit (Zaken et al., 2021), Adapters (Houlsby et al., 2019), Prompt Tun-

ing (Liu et al., 2023b; 2022; Lester et al., 2021), Prefix Tuning (Li & Liang, 2021) and Low-Rank Adaptation (LoRA) (Hu et al., 2022). Among these methods, LoRA is the most representative one. It introduces trainable low-rank matrices to approximate weight updates, realizing highly efficient fine-tuning with low cost, which has led to various extensions (Kopiczko et al., 2024; Liu et al., 2024; Valipour et al., 2023; Zhang et al., 2023a; Ding et al., 2023). For example, VeRA (Kopiczko et al., 2024) further reduced the number of trainable parameters in LoRA by employing shared low-rank matrices and trainable scaling vectors. DoRA (Liu et al., 2024) enhanced fine-tuning performance and stability by decomposing the pre-trained weights into magnitude and direction components. For greater flexibility in LoRA's rank, DyLoRA (Valipour et al., 2023) dynamically trained all ranks during training to avoid separate rank tuning for each task. AdaLoRA (Zhang et al., 2023a) allocated the parameter budget based on the importance scores of the weight matrices and pruned insignificant singular values to exclude unimportant rank spaces. SoRA (Ding et al., 2023) introduced a trainable gating unit and used proximal gradient descent to optimize the sparsity of the update matrices, dynamically adjusting the intrinsic rank size during training.

However, LoRA's fixed-rank constraint limits its flexibility. Although recent works (Valipour et al., 2023; Zhang et al., 2023a) have enhanced LoRA's adaptability, they predominantly address single-task training scenarios. These approaches do not consider multi-task scenarios, where selecting the most suitable rank for different tasks remains an open challenge. This gap underscores the need for more flexible and adaptive methods capable of efficiently handling diverse and concurrent tasks in multi-task learning scenarios.

### 2.2 MULTI-TASK LEARNING

Multi-task learning (MTL) focuses on simultaneously solving multiple related tasks with a single model, which has been studied from multiple perspectives and offers several advantages (Zhang & Yang, 2021; Vandenhende et al., 2022). As large language models advance, multi-task learning has become an essential skill for them. However, it still faces several challenges, such as conflicts between different tasks, balancing task weights, and the demands on training resources (Chen et al., 2021; Kollias et al., 2024). Optimizing PEFT methods for MTL scenarios is a highly valuable direction. For instance, HyperFormer (Mahabadi et al., 2021) improved Adapter-based methods by using a shared hypernetwork for cross-task knowledge sharing while integrating task-specific Adapters. In Prompt Tuning, SPoT (Vu et al., 2022) learned prompts from source tasks and adapted them for target tasks, enhancing model performance. ATTEMPT (Asai et al., 2022) used an attention mechanism to merge source and target prompts for effective knowledge transfer. MPT (Wang et al., 2023b) employed prompt decomposition and knowledge distillation to create a transferable prompt, which was then fine-tuned with low-rank modifications for specific tasks.

Additionally, LoRA-based improvements have shown significance in multi-task scenarios. Multi-LoRA (Wang et al., 2023a) employed multiple parallel LoRA modules during training, ensuring that the rank space closely approximates fine-tuning. MixLoRA (Li et al., 2024) used multiple parallel LoRA experts with a gating mechanism to select the appropriate expert for each token. MOELoRA (Liu et al., 2023a) learned task-shared and specific knowledge with multiple experts and adjusted their contributions for each task using a gating function. However, these methods also have limitations. The parallel LoRA modules increase the number of trainable parameters, significantly reducing training efficiency. Besides, they do not account for the varying rank requirements of different tasks.

## 3 PRELIMINARY

### 3.1 PROBLEM DEFINITION

In multi-task learning scenarios, the objective is to concurrently learn multiple tasks, each characterized by potentially diverse data distributions and goals. Formally, we consider a set of tasks $T = \{\mathcal{T}_1, \mathcal{T}_2, \ldots, \mathcal{T}_T\}$, where each task $\mathcal{T}_t$ is associated with a dataset $\mathcal{D}_t = \{(x_i^t, y_i^t)\}_{i=1}^{N_t}$ comprising $N_t$ input-output pairs. Here, $x_i^t$ denotes the input data and $y_i^t$ denotes the corresponding label or output for task $\mathcal{T}_t$. The target is to learn a shared model $F$ with parameters $\boldsymbol{\theta}$ to satisfy the requirements of different simultaneously.

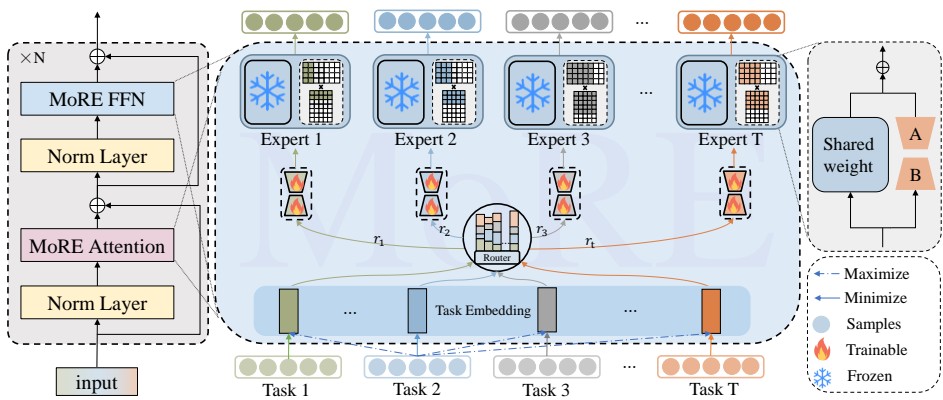

Figure 1: The overall framework of our proposed MoRE.

## 3.2 LoRA: Low-Rank Adaptation

(LoRA) (Hu et al., 2022) is designed to reduce the computational cost and memory footprint of adapting LLMs to new tasks. Instead of fine-tuning all parameters, LoRA adapts the model by introducing low-rank updates to the existing weight matrices. Formally, let $\boldsymbol{W}_0 \in \mathbb{R}^{m \times d}$ be a weight matrix of a specific LLM, where $m$ and $d$ denote the input and output dimensions, respectively. LoRA approximates the weight update $\Delta \boldsymbol{W}$ using a low-rank decomposition:

$$\Delta \boldsymbol{W} = \boldsymbol{B}\boldsymbol{A}, \tag{1}$$

where $\boldsymbol{A} \in \mathbb{R}^{r \times d}$ and $\boldsymbol{B} \in \mathbb{R}^{m \times r}$ are the learned low-rank matrices, and $r \ll min(m, d)$ is the pre-defined rank. This decomposition significantly reduces the number of trainable parameters from $m \times d$ to $r \times (m + d)$. For original $\boldsymbol{h} = \boldsymbol{W}_0\boldsymbol{x}$, the modified forward pass yields:

$$\boldsymbol{h} = \boldsymbol{W}_0\boldsymbol{x} + \Delta \boldsymbol{W}\boldsymbol{x} = \boldsymbol{W}\boldsymbol{x} + \boldsymbol{B}\boldsymbol{A}\boldsymbol{x}. \tag{2}$$

However, this process highly depends on the pre-defined rank $r$, which is time-consuming and computationally expensive to search. And this problem will be amplified in multi-task scenarios, limiting the potential of LoRA. Thus, *How to use LoRA to achieve efficient LLM fine-tuning in multi-task scenarios* is the main focus of our paper.

## 4 Mixture of Low-Rank Experts

To tackle the inefficient problem of LoRA in multi-task scenarios, we propose a novel Mixture of Low-Rank Experts (MoRE). The cores lie in *how to learn experts* and *how to select them*. As illustrated in Figure 1, we focus on parameters in attention layer and FFN layer of the Transformer block. We first assign a task embedding for each task to describe the abstract task characteristics. Then, based on the task embedding, we design a novel *adaptive rank selector* to select the appropriate rank for each task, term as the rank expert. Finally, we incorporate contrastive learning to ensure the quality of learned task embedding and design a *Balanced Data Sampling strategy* to stabilize the learning process for better multi-task learning. Next, we will introduce each part in detail.

## 4.1 Task Embedding

Existing multi-task learning methods focus on mining useful information from task data and transferring knowledge from one task to another. Despite the progress, they are still weak at sharing common information among tasks and distinguishing specific information aligning with each task. This shortcoming will prohibit the efficiency of PEFT methods when using them to tune LLMs in multi-task scenarios. Therefore, we propose using task embeddings to represent different tasks so that task characteristics can be summarized comprehensively. This operation is also the precondition of our designed rank expert for measuring the connections and distinctions among different tasks.

Specifically, we use matrix $\boldsymbol{E} = \{\boldsymbol{e}_1, \boldsymbol{e}_2, ..., \boldsymbol{e}_l\}$ to denote all tasks, where $\boldsymbol{e}_i$ represents the $i^{th}$ task in the multi-task scenarios. Then, we leverage Kaiming Initialization to initialize them and learn precise $\boldsymbol{E}$ during model training. Since there is no supervised signal for $\boldsymbol{E}$, we design a Contrastive Learning (CL) based optimization target to learn them, which will be introduced in Section 4.3.

## 4.2 ADAPTIVE RANK SELECTOR

As illustrated in Section 1, vanilla LoRA and its typical variances usually have a fixed rank $r$, which is pre-defined by experts. However, different tasks may benefit from different ranks depending on their complexity and data distributions (Valipour et al., 2023; Ding et al., 2023). Searching the best rank is time-consuming and computationally expensive. Meanwhile, training parallel LoRA modules or multiple LoRAs when applying LLMs to multi-task scenarios will amplify the problem and prohibit the effectiveness, causing high computational and storage costs. Therefore, we employ Mixture-of-Experts (MoE) framework and design a novel *Adaptive Rank Selector*.

Different from previous work that treated the entire LoRA module as an expert, we propose to treat the rank $r$ as the expert and use one LoRA to realize LLM fine-tuning in multi-task scenarios. Assuming the selected rank of LoRA is $r$, the rank expert can be selected within the range $[1, r]$. *Along this line, different experts can share common information at the overlap part in the learned metrics* (i.e., $A$ and $B$) *and align specific information corresponding to each task at the non-overlap part.* Formally, we use the learned task embedding $e_t$ to select the appropriate rank from the LoRA module and leverage a gating network $G(\cdot)$ to guarantee the quality of the selection. Let $\{1, 2, \ldots, r\}$ be the set of experts' ranks. For task $\mathcal{T}_t$, $G(\cdot)$ takes $\mathbf{e}_t$ as input and outputs a probability distribution over rank experts as follows:

$$\mathbf{p}_t = G(\mathbf{e}_t) = \mathrm{softmax}(\mathbf{W}_g \mathbf{e}_t + \mathbf{b}_g), \qquad (3)$$

where $\{\mathbf{W}_g, \mathbf{b}_g\}$ are learnable parameters. The probability distribution $\mathbf{p}_t \in \mathbb{R}^r$ indicates the relevance of each rank to task $\mathcal{T}_t$. During the forward pass, we select the rank with the highest probability and use the selected rank to truncate the LoRA module for rank expert construction. Then, MoRE uses LoRA paradigm to realize the fine-tuning as follows:

$$r_t = \arg\max \mathbf{p}_t,$$
$$\boldsymbol{h} = \mathbf{W_0}x + \mathbf{B_t}\mathbf{A_t}x, \quad \mathbf{A_t} = \mathbf{A}[: r_t, :], \quad \mathbf{B_t} = \mathbf{B}[:, : r_t]. \qquad (4)$$

One step further, during the backward pass, the $\arg\max$ in Eq.(3) is non-differentiable, causing $G(\cdot)$ unable to be learned. Thus, we incorporate Straight-Through Estimator (STE) (Bengio et al., 2013) technique to address this issue. Specifically, we use STE to calculate the approximate gradient to allow the gradient to propagate back to $G(\cdot)$ correctly:

$$\mathrm{Ste}(\mathbf{p}_t) = \mathbf{p}_t + sg[one\_hot(\mathbf{p}_t) - \mathbf{p}_t], \qquad (5)$$

where $one\_hot(\cdot)$ is a function that converts a vector into its one-hot version. $sg(\cdot)$ stands for stop gradient. Then, we modify the forward process in Eq.(4) as:

$$\mathbf{h} = \mathbf{W_0}x + \mathrm{Ste}(\mathbf{p}_t)[r_t] \cdot \mathbf{B_t}\mathbf{A_t}x. \qquad (6)$$

Thus, Adaptive Rank Selector module can realize a precise selection of rank experts. Furthermore, since MoRE uses the overlap part among LoRA metrics to share the common information across different tasks, the lower part will be updated more frequently during fine-tuning. Thus, its learning rate should be small for a slow and stable updating. To realize this goal, we perform a linear scaling on its weights for the balance:

$$\mathbf{h} = \mathbf{W_0}x + \mathrm{Ste}(\mathbf{p}_t)[r_t] \cdot \frac{r_t}{|T|}\mathbf{B_t}\mathbf{A_t}x, \qquad (7)$$

where $|T|$ represents the total number of tasks. To verify the effectiveness of this design, we also conducted an ablation study on this operation in Section 5.4.

## 4.3 BALANCED DATA SAMPLING AND CL-BASED OPTIMIZATION

**Balanced Data Sampling.** In multi-task scenarios, data distributions of different tasks are also essential for LLM fine-tuning. For instance, in GLUE benchmark (Wang et al., 2018), MNLI and RTE datasets have proportionally disparate data distributions (i.e., $392,000$ v.s. $2,500$ examples). If this attribute is not considered when fine-tuning LLMs in multi-task scenarios, it is obvious that fine-tuned LLMs will underfit the task with smaller datasets.

In response, we propose a simple but effective Balanced Dataset Sampling strategy to ensure each dataset contributes proportionally during the fine-tuning process, regardless of its size. Specifically, we assign a sampling weight $\phi_t$ to each dataset $\mathcal{D}_t$, which is inversely proportional to its size:

$$\Phi = [\phi_1, \phi_2, ..., \phi_T], \quad \phi_t = \exp\left(\frac{|\mathcal{D}_t|}{\sum_{i=1}^{T} |\mathcal{D}_i|}\right), \tag{8}$$

$$D_t = Sampling(D, \Phi),$$

where $|\mathcal{D}_t|$ is the size of dataset $\mathcal{D}_t$. $Sampling(D, \Phi)$ denotes sampling a subset from all datasets $D$ with the probability distribution $\Phi$. This dynamic sampling strategy helps to balance the contributions of different datasets, thereby reducing the risk of underfitting smaller datasets and improving the overall performance of the multi-task training.

**CL-based Optimization.** As mentioned in Section 4.1, there is no supervised signal for task embedding learning. Thus, one important question should be considered: "*How to ensure the task characteristics and task distinguishability of the learned task embedding without annotation requirements?*" In response, we propose to leverage CL to ensure the quality of learned task embeddings. Consider a batch $\mathcal{B}$ of samples, where all samples in $\mathcal{B}$ belong to the same task $\mathcal{T}_t$. Let $\{\mathbf{x}_i\}_{i=1}^{N}$ be the set of $N$ samples in $\mathcal{B}$, and let $\mathbf{h}_i$ be the representation of sample $\mathbf{x}_i$ obtained from the model. The task embedding for task $\mathcal{T}_t$ is denoted as $\mathbf{e}_t$. The optimization target can be formulated as follows:

$$\mathcal{L}_{con} = \frac{1}{N} \sum_{i=1}^{N} \left[ \log \frac{\exp\left(\frac{sim(\mathbf{h}_i, \mathbf{e}_t)}{\tau}\right)}{\sum_{k=1}^{T} \exp\left(\frac{sim(\mathbf{h}_i, \mathbf{e}_k)}{\tau}\right)} \right], \tag{9}$$

where $sim(\cdot, \cdot)$ denotes a similarity measure, such as the dot product or cosine similarity, and $T$ is the total number of tasks. $\tau$ is the temperature. $\mathbf{e}_t$ and $\mathbf{e}_k$ are the $t^{th}$ and $k^{th}$ tasks where $t \neq k$. By using Eq.(9), we can measure the connection between task embedding $\mathbf{e}_t$ and its data samples $\{\mathbf{x}_i\}_{i=1}^{N}$. Since each data sample is close to the corresponding task embedding, we can conclude the learned task embeddings can be used to describe task characteristics, which is also supported by experimental results in Section 5.3.

Besides using contrastive loss to learn task embeddings, we also select generation loss $\mathcal{L}_{gen}$ to measure the discrepancy between the sequences generated by the model and the target sequences. Let $\boldsymbol{y}$ and $\hat{\boldsymbol{y}}$ be target sequence and generation, $\mathcal{L}_{gen}$ can be formulated with the cross-entropy loss:

$$\mathcal{L}_{gen} = -\sum_{t=1}^{T} y_t \log \hat{y}_t. \tag{10}$$

Then, we leverage a hyperparameter $\lambda$ to balance the contributions of the generation loss and the contrastive loss, and formulate the overall optimization target of MoRE as follows:

$$\mathcal{L} = \mathcal{L}_{gen} + \lambda \mathcal{L}_{con}. \tag{11}$$

**Discussion.** Compared with existing LoRA-based PEFT methods and MoE-based fine-tuning methods, our proposed MoRE has the following properties. 1) We propose to treat different rank values in one LoRA module as experts, and design an adaptive rank selector to select appropriate rank experts for different tasks, which can effectively share the common information among tasks and emphasize the specific information aligned to each task; 2) We leverage task embeddings to accurately describe the abstract task characteristics with a CL optimization; 3) We also consider task data distributions and design a simple but effective Balanced Data Sampling strategy to ensure the capability of fine-tuned LLMs on different tasks.

## 5 EXPERIMENTS

### 5.1 EXPERIMENTAL SETUP

**Datasets.** We utilized GLUE benchmark (Wang et al., 2018) to evaluate the model performance. GLUE covers multiple tasks of paraphrase detection (MRPC, QQP), sentiment classification (SST-2), natural language inference (MNLI, RTE, QNLI), and linguistic acceptability (CoLA). Following previous work (Zhang et al., 2021), for those datasets with fewer than $10,000$ samples (i.e., RTE, MRPC, STS-B, CoLA), we split the original validation set into new validation and test sets equally.

Table 2: Performance on GLUE benchmark. For STS-B, we report Pearson correlation coefficients. For CoLA, we report Matthews correlation. For all other tasks, we report accuracy. **Bold** and underlined fonts indicate the best and the second-best results.

| Methods | params/task | MNLI | QQP | QNLI | SST-2 | STS-B | MRPC | RTE | CoLA | AVG |
|---|---|---|---|---|---|---|---|---|---|---|
| Finetuning | 28M | 85.7 | **91.1** | 92.0 | 92.5 | 88.8 | 90.2 | 75.4 | 54.9 | 83.8 |
| Adapters | 1.8M | **86.3** | 90.5 | 93.2 | 93.0 | 89.9 | 90.2 | 70.3 | 61.5 | 84.4 |
| PT | 9.6k | 85.6 | 90.6 | 93.2 | 93.9 | 89.9 | 86.3 | 67.6 | 55.3 | 82.8 |
| $LoRA_{r=8}$ | 0.39M | 85.8 | 89.2 | 93.1 | 93.2 | 90.4 | 89.9 | 76.3 | 62.8 | 85.1 |
| $LoRA_{r=16}$ | 0.78M | 84.9 | 89.6 | 93.0 | 93.7 | 90.4 | 88.7 | 80.6 | 63.9 | 85.6 |
| HyperFomer | 638K | 85.7 | 90.0 | 93.0 | 94.0 | 89.7 | 87.2 | 75.4 | 63.7 | 84.8 |
| MPT | 10.5K | 84.3 | 90.0 | 93.0 | 93.3 | 90.4 | 89.2 | 82.7 | 63.5 | 85.8 |
| MultiLoRA | 1.56M | 85.9 | 89.7 | 92.8 | **94.5** | 89.8 | 88.2 | 80.6 | 66.9 | 86.0 |
| MixLoRA | 1.49M | 85.8 | 90.0 | 92.9 | 93.7 | 90.3 | 89.2 | 78.4 | 67.2 | 85.9 |
| MOELoRA | 0.78M | **86.3** | 90.1 | 93.2 | 94.2 | 90.0 | 89.7 | 81.3 | 68.4 | 86.7 |
| MoRE | 0.78M | 86.2 | 90.0 | **93.4** | 93.7 | **90.7** | **91.2** | **83.5** | **69.9** | **87.3** |
| LLaMA2-LoRA | 2.5M | 86.9 | 88.6 | 93.5 | 96.2 | 90.2 | **92.6** | 89.2 | 65.0 | 87.8 |
| LLaMA2-MultiLoRA | 10M | 87.6 | 85.0 | 93.4 | 96.7 | 92.2 | 88.7 | 87.8 | 72.4 | 88.0 |
| LLaMA2-MixLoRA | 12.2M | 86.8 | 88.1 | 93.6 | 96.0 | 91.3 | 88.2 | 87.1 | **73.2** | 88.0 |
| LLaMA2-MOELoRA | 5M | 87.0 | 87.6 | 91.4 | 96.3 | **92.4** | 91.2 | 87.8 | 64.4 | 87.3 |
| LLaMA2-MoRE | 5M | **89.4** | **89.0** | **94.4** | **96.9** | 92.2 | 89.2 | **92.1** | 66.9 | **88.8** |

For others, we randomly select $1,000$ examples from training set as the validation set, and use original validation sets as test sets. Additionally, we included the BoolQ (Clark et al., 2019), PIQA (Bisk et al., 2020), OBQA (Mihaylov et al., 2018), and ARC (Clark et al., 2018) datasets to assess the model's performance in commonsense reasoning tasks. These datasets provide a variety of challenges that require understanding of everyday scenarios and logical reasoning. Moreover, we select SciTail (Khot et al., 2018), BoolQ (Clark et al., 2019), and CB (de Marneffe et al., 2019) datasets to evaluate model robustness and generalization capabilities in few-shot learning scenarios. We also provide an early attempt on generation tasks (Nan et al., 2021) and report results in Appendix A.

**Baselines.** We compare MoRE with the following baselines: (1) Full fine-tuning (FT), (2) Vanilla Adapter (Houlsby et al., 2019), (3) Vanilla prompt tuning (PT) (Raffel et al., 2020), (4) Vanilla LoRA (Hu et al., 2022), We also select the following advanced multi-task PEFT baselines: (1) HyperFomer (Mahabadi et al., 2021), (2) MPT (Wang et al., 2023b), (3) MultiLoRA (Wang et al., 2023a), (4) MixLoRA (Li et al., 2024). (5) MOELoRA (Liu et al., 2023a). All methods are tuned based on reported settings with T5-base and LLaMA2-7B as backbone for a fair comparison.

**Training Setup.** We select T5-base (Raffel et al., 2020) and LLaMA2-7B (Touvron et al., 2023) as the backbone. The optimizer is AdamW. The learning rate is $3 \times 10^{-4}$, with a linear decay schedule and warm-up over the first 500 steps. The batch size is 32, and the training process spanned 5 epochs. The maximum input sequence length is 128 tokens. The $\lambda$ is set to 0.1, and the temperature $\tau$ in Eq.(9) is 0.05. For few-shot domain transfer, We use the best checkpoint trained on the GLUE tasks for initialization. The task embeddings of the most similar tasks will be shared (e.g., MNLI for SciTail and CB). The T5-base is trained on Ubuntu 20.04 platform using two NVIDIA RTX 4090 GPUs, and LLaMA2-7B is trained with four NVIDIA Tesla A100 PCIe GPUs.

**Evaluation Setup.** For GLUE benchmark and commonsense reasoning tasks, we selected the checkpoint with the highest average performance on validation set. For few-shot learning, we performed training and testing under each shot setting using 5 random seeds. Then, we reported the average performance for a fair and robust estimation and comparison.

## 5.2 OVERALL PERFORMANCE

**Performance on GLUE Benchmark and Commonsense Reasoning** Tables 2 and 3 present model performance on multi-task scenarios. From Table 2, we can observe that MoRE achieves impressive performance over different tasks with a relatively small number of fine-tuned parameters. Moreover, compared with LoRA implementation ($r = 16$), MoRE achieves significant improvement (e.g., 1.7% average improvement) without extra tuning parameter, demonstrate the superiority of MoRE.

Table 3: Performance (Accuracy) of all methods on Commonsense Reasoning scenarios.

| Methods | params/task | BoolQ | PIQA | OBQA | ARC-E | ARC-C | AVG |
|---|---|---|---|---|---|---|---|
| LLaMA2-LoRA | 2.5M | 80.9 | 77.7 | 79.0 | 83.7 | 76.9 | 79.6 |
| LLaMA2-MultiLoRA | 10M | 76.5 | 72.9 | 68.2 | 81.6 | 61.9 | 72.2 |
| LLaMA2-MixLoRA | 12.2M | 84.3 | 79.5 | 82.6 | **86.8** | 76.3 | 81.9 |
| LLaMA2-MOELoRA | 4.5M | 84.0 | 79.9 | 81.8 | **86.8** | **77.3** | 82.0 |
| LLaMA2-MoRE | 4.5M | **87.2** | **82.3** | **83.0** | 86.7 | 74.2 | **82.7** |

By treating ranks as experts and learning accurate task embedding to support the expert selection, MoRE can effectively share common information and specify aligned information across different tasks, and achieve impressive performance without too many fine-tuned parameters. Furthermore, the number of fine-tuned parameters of MoRE is the same as representative $LoRA_{r=16}$, smaller than other LoRA-based improvement methods. When fine-tuning larger models (e.g., LLaMA2-7B), MoRE can achieve much better performance. All these phenomena demonstrate the superiority of MoRE. Additionally, MoRE can be further optimized during inference, allowing the inference-time parameters to be comparable to those of $LoRA_{r=8}$ (see Section 5.4 for details).

For PEFT baselines, we observe that fine-tuned performance over small datasets (i.e., MRPC, RTE, and CoLA) is not so good. One possible reason is that they do not consider the shared knowledge across different tasks, treating each task individually. Moreover, their fine-tuned module requires more training data. Thus, we can observe the sub-optimal multi-task performance on these baselines. For multi-task baselines, though they consider the shared knowledge across different tasks, they do not distinguish the distinctions and connections among different tasks. For example, HyperFormer just learns task embeddings without any constraints. MPT sacrifices too much on MNLI task. Therefore, their performance is not comparable with MoRE. As for MultiLoRA and MixLoRA, although they improve the performance of LoRA, they do not utilize task-aware modules or consider specific rank allocation for different tasks. As a result, their performance improvements are limited.

Meanwhile, we can obtain the similar results from Table 3. MoRE achieves the highest average accuracy when fine-tuning LLaMA2 on commonsense reasoning scenario, surpassing all other approaches with a smaller number of fine-tuned parameters, proving the superiority of MoRE. In contrast, MultiLoRA has worse performance. We speculate the reason is that commonsense reasoning tasks require more fine-grained task information sharing and distinguishing, which cannot be satisfied by a simple LoRA ensemble.

**Performance on Few-shot Domain Transfer** To further verify the efficiency of MoRE, we conduct few-shot domain transfer experiments and report results in Table 4. We can observe MoRE achieves stable and optimal performance over most datasets with different few-shot settings, This consistency proves the efficiency of MoRE on sharing common information and distinguishing specific information across different tasks, which helps leverage minimal data for effective transfer learning.

For baselines, traditional fine-tuning and multi-task tuning (i.e., HyperFomer and MPT) require more parameters to be tuned. Thus, their performance is subpar with limited training data. Increasing training data will improve their performance. For LoRA-based baselines, multi-task LoRA-based methods have similar performance with vanilla $LoRA_{r=16}$ and do not show performance gains in multi-task learning. We speculate that they may encounter difficulties in efficiently allocating appropriate ranks or adapting parameters to new tasks when only a small number of samples is available. These phenomena highlight the challenges of effectively utilizing few-shot data to achieve good generalization across different domains, demonstrating the superiority of MoRE.

### 5.3 EXPERT SELECTION ANALYSIS AND TASK VISUALIZATION

**Low-Rank Expert Allocation.** To investigate the expert distribution after model fine-tuning, we analyze the expert allocation across all layers for each task. As shown in Figure 2(a), in most cases, all tasks heavily rely on experts with ranks 1, 2, or 3. This indicates a significant amount of parameter redundancy in LoRA, where the higher-rank parameters in many modules do not substantially contribute during the fine-tuning process. This is consistent with our design that MoRE leverage rank experts to share common information across different tasks with lower-rank parameters. Moreover, to illustrate the dependency of different tasks on various ranks, and to eliminate the influence

Table 4: Few-shot domain transfer results (Accuracy) of T5-base models fine-tuned on GLUE averaged across 5 seeds. **Bold** and underlined fonts indicate the best and the second-best results.

| Task | k-shot | Finetuning | LoRA | HyperFomer | MPT | MultiLoRA | MixLoRA | MOELoRA | MoRE |
|------|--------|-----------|------|-----------|-----|-----------|---------|---------|------|
| BoolQ | 4 | 50.5 | 64.2 | 48.0 | 62.2 | **65.2** | 62.8 | 64.0 | 64.6 |
| | 16 | 56.5 | 66.1 | 50.2 | 63.3 | 65.8 | 64.4 | 64.8 | **66.2** |
| | 32 | 58.4 | 67.4 | 58.3 | **68.9** | 67.6 | 66.2 | 65.7 | 67.9 |
| CB | 4 | 57.7 | 84.3 | 60.7 | 73.6 | 85.0 | **86.6** | 85.4 | 85.7 |
| | 16 | 77.0 | 85.7 | 76.3 | 78.6 | 85.7 | **86.4** | 86.3 | **86.4** |
| | 32 | 80.0 | 87.1 | 81.4 | 82.1 | 86.6 | **89.3** | 88.3 | 88.6 |
| SciTail | 4 | 79.6 | 80.8 | 82.0 | 80.2 | 78.1 | 77.5 | 80.4 | **83.8** |
| | 16 | 80.0 | 84.0 | 86.5 | **87.3** | 81.7 | 82.4 | 83.1 | 86.7 |
| | 32 | 81.9 | 85.3 | 85.8 | 86.3 | 83.6 | 83.3 | 84.5 | **87.4** |

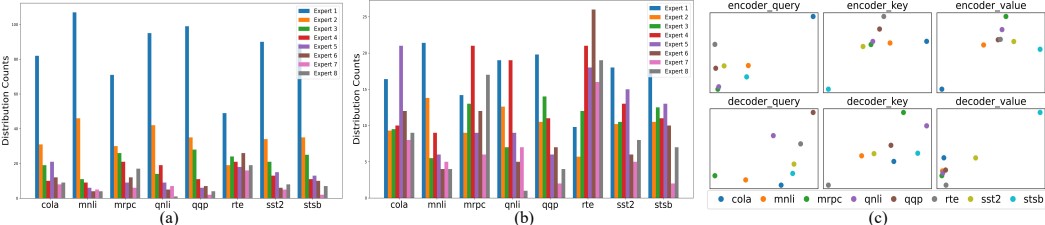

Figure 2: (a)-(b) The distribution of expert allocation. (c) Visualization of the task embeddings.

of low-rank experts 1-3, we proportionally scaled the low-rank experts 1-3 for each task. As depicted in Figure 2(b), different tasks indeed exhibit stronger dependencies on different ranked experts. For instance, MRPC relies on expert 4, RTE depends on expert 6, and STSB still relies on expert 1. This phenomenon proves strong evidence to support that MoRE can allocate more suitable ranks for different tasks and use higher-rank parameters to emphasize the specific information aligned to each task. This is also the reason that MoRE can achieve the best performance in multi-task scenarios.

**Visualization of Task Embeddings.** In Section 4.1, we mention that task embedding is essential for rank expert selection. Here, we visualize the learned task embeddings to verify the quality and provide some insight for understanding MoRE. Specifically, we use PCA to process the task embeddings from the final layer of the self-attention module and report results in Figure 2(c). We can observe that task embeddings exhibit varying degrees of clustering. Similar tasks (e.g., MRPC and QNLI) have a tendency to cluster, while different tasks have a large distance. Moreover, STSB and CoLA tasks seem relatively independent to all other tasks. It is consistent with our expectations. Since STSB involves similarity computation and other tasks are classification tasks, they indeed have obvious differences in abstract task characteristics. These phenomena also give us some insight into the efficiency and effectiveness of MoRE. By using task embeddings to extract abstract task information, MoRE can capture and represent the similarities and differences between tasks in diverse ways. This capability helps to select the appropriate rank expert, which in turn ensures the superiority of our proposed MoRE. We also provide more examples in Appendix C.

## 5.4 ABLATION STUDY AND PARAMETER ANALYSIS

**Ablation Study.** We perform an ablation study to better verify the contribution of each component in MoRE. As shown in Table 5, we can observe a significant performance decrease (i.e., $0.9\%$ and $0.7\%$ average accuracy decrease) when replacing task-specific embeddings with a shared embedding (w/o Task Embeddings) or removing contrastive optimization (Eq.(9)) (w/o CL optimization), proving the importance of the CL-based task embedding module. This is consistent with our design in Section 4.1. Moreover, removing STE and using soft expert selection with Eq.(3) (w/o STE) also have a big impact on the model performance. Since we treat different ranks as experts, using soft expert selection will reduce the discrimination between different experts, leading to a tendency towards vanilla $LoRA_{r=16}$. Furthermore, when using random sampling (w/ Random Sample), we can observe $86.2\%$ average GLUE accuracy, indicating the positive impact of our balanced sample selection strategy. Besides, when removing linear scaling (w/o Linear Scaling), we observed a slight drop in performance, indicating that this adjustment helps mitigate the overfitting of LoRA.

Table 5: Ablation study results (Average Results on GLUE benchmark) of MoRE.

| Conditions | GLUE Avg. |
|---|---|
| MoRE | **87.3** |
| w/o Linear Scaling | 87.0 |
| w/o Task Embeddings | 86.1 |
| w/o CL optimization | 86.3 |
| w/o STE | 86.4 |
| w/ Random Sample | 86.2 |

| Method | Parameter |
|---|---|
| LoRA | $6Lr(m + d)$ |
| MultiLoRA | $6nLr(m + d) + 6Ld$ |
| MixLoRA | $2nLr(m + d) + 2Lnm$ |
| MOELoRA | $6Lr(m + d) + 6Lh(n + T)$ |
| MoRE | $6Lr(m + d) + 6Lh(r + T)$ |

(a)

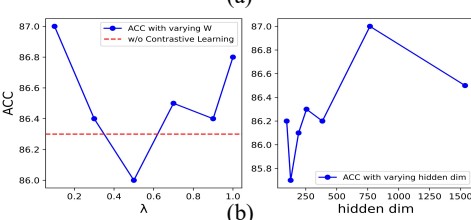

(b)

Figure 3: (a) Comparison of trainable parameters. (b) Parameter sensitivity experiments.

**Parameter Sensitivity Test.** There are two hyper-parameters that affect model performance: (1) $\lambda$ in Eq.(11); (2) the dimension of task embeddings. Therefore, we conduct additional experiments to verify their impacts and report results in Figure 3(b). From the figure, we have the following observations. First, with the increase of $\lambda$, model performance first decreases and then increases. Since different tasks have various data samples, the corresponding contrastive loss would exhibit oscillations, which may have negative impacts on fine-tuning performance. Thus, we set $\lambda = 0.1$ to obtain the best performance. Second, with the increase of the dimension of task embedding, model performance will first increase and then decrease. We attribute this phenomenon to the following reasons. When the dimension is too small, task embedding cannot capture complex task information, which will harm the selection of rank experts. When the dimension is too big, its training requires more data, which cannot always be satisfied. Moreover, since we need to calculate similarity with samples, the sample embedding from LLMs also should be considered. Based on all these reasons, we finally set the dimension of task embeddings as 768.

**Parameter Efficiency.** To analyze the model complexity, we give the number of tuning parameters of different LoRA-based methods, which is summarized in Figure 3 (a). The notation explanations are as follows: $\{L, r, (m, d), n, T, h\}$ refer to model layers, LoRA rank, model dimensions, parallel LoRA module number, task number, and task embedding dimension. Compared with tuning parameter size of LoRA, the added parameter number of MoRE is $6Lh(r + T)$, including the extra task embeddings ($Th$ for a single LoRA module) and adaptive rank selector ($rh$ for a single LoRA module). Compared with MultiLoRA and MixLoRA which use parallel module design to tackle multi-task learning, MoRE is more efficient. Moreover, once our task embedding and gate modules are trained, we can construct a mapping from tasks to experts, which allows us to avoid the repeated computation of the task embedding and gate modules during inference, thereby reducing the parameter count to be consistent with $LoRA$. This is also the reason why MoRE achieves impressive performance in multi-task scenarios without too many fine-tuning parameters. We also provide detailed parameter size calculations of baselines in Appendix B.

## 6 CONCLUSION AND FUTURE WORK

In this paper, we argued that existing PEFT methods either did not consider multi-task fine-tuning or used parallel structures that added too many tuning parameters, prohibiting the efficiency of LoRA. In response, we proposed a novel MoRE for multi-task PEFT. By treating each low-rank in LoRA module as a specialized expert, MoRE could share common information with lower-rank parameters and emphasize the specific information aligned to each task with higher-rank parameters, which not only fully exploited the potential of LoRA module but also reduced the tuning parameter size in multi-task scenarios. To ensure the quality of rank experts, we used task embeddings to capture the distinctions and connections among different tasks. We also developed a CL-based optimization target and Balanced Dataset Sampling strategy to ensure the fine-tuning quality. Extensive experiments demonstrated that our method achieves significant improvements on the GLUE benchmark and exhibits strong transfer learning performance. In the future, we plan to extend the application of MoRE and design a more efficient module to further improve its capability.

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

Table 6: Model performance on NLG tasks

| Method | DART | E2E | WebNLG | AVG |
|--------|------|-----|--------|-----|
| FT | **46.1** | 61.4 | 44.2 | **50.6** |
| **LoRA**$_{r=8}$ | 43.2 | 60.6 | 43.8 | 49.2 |
| **LoRA**$_{r=16}$ | 44.6 | 60.8 | 44.3 | 49.9 |
| MultiLoRA | 44.0 | 61.3 | 44.9 | 50.1 |
| MixLoRA | 44.3 | 60.9 | **45.3** | 50.2 |
| MoRE | 45.0 | **61.5** | 45.1 | 50.5 |

## A  ADDITIONAL EXPERIMENTS ON NLG

To further validate the effectiveness of our method, we conducted experiments on natural language generation (NLG) tasks using three datasets: DART, E2E, and WebNLG. DART focuses on generating text from structured data, E2E involves generating restaurant descriptions from key attributes, and WebNLG is designed for generating text from knowledge graph triples. As shown in Table 6, none of the methods outperform fine-tuning (FT) on NLG tasks, and LoRA shows a significant performance drop. This indicates that using a fixed rank for training all tasks is suboptimal. In contrast, our method achieves performance comparable to FT. This is attributed to our method's ability to allocate an appropriate rank for different tasks efficiently.

## B  DETAILED CALCULATION OF PARAMETER COUNTS

**LoRA parameters:**  LoRA employs matraix A and B to introduce low-rank adaptations in both the attention layers (q, k, v, o) and the feed-forward network (FFN) layers ($w_i$, $w_o$) of the T5-base model. Each LoRA layer has $r(m + d)$ paramters. The total number of parameters for LoRA with L transformer layers is $6Lr(m + d)$.

**MultiLoRA parameters:**  MultiLoRA employs parallel LoRA models for training, so its parameter count is $n$ times that of vanilla LoRA, where $n$ is the number of parallel LoRA modules. Additionally, MultiLoRA modifies the scaling factors to be learnable parameters (with parameter count $d$). Therefore, the total number of parameters is $6nLr(m + d) + 6Ld$.

**MixLoRA parameters:**  MixLoRA only employs parallel expert LoRA modules in the FFN layers and uses a gating module (with parameter count $nm$) to select the appropriate LoRA expert. Therefore, the total number of parameters is $2nLr(m + d) + 2Lnm$.

**MOELoRA Parameters:**  MOELoRA utilizes parallel LoRA models with a rank of $r/n$ and incorporates a task embedding module to represent each task (with a parameter count of $Th$). Additionally, it employs a gating module (with a parameter count of $nh$) to compute the weights for each LoRA. Therefore, the total number of parameters is given by $6Lr(m + d) + 6Lh(n + T)$.

**MoRE parameters:**  Our proposed MoRE employs the same LoRA modules as vanilla LoRA, but treats LoRA modules with different ranks as experts, thereby introducing an additional gating module (with parameter count $rh$). To better adapt to different tasks, we also introduce a task embedding module (with parameter count $Th$, where h is the hidden dimension). Therefore, the total number of parameters is $6Lr(m + d) + 6Lh(r + T)$. In the GLUE dataset, $T = 8$ is consistent with $r = 8$. If the hidden dimension is set to be the same as $d$, then the parameter count is $12Lr(m + d)$, which is exactly the same as the parameter count with $LoRA_{r=16}$. Compared to MultiLoRA and MixLoRA, we do not use a parallel module design, so there is no parameter $n$ that leads to a parameter count far exceeding that of LoRA. Furthermore, once our task embedding and gate modules are trained, we can construct a mapping from tasks to experts. This allows us to avoid the repeated computation of the task embedding and gate modules during inference, thereby reducing the parameter count to be consistent with $LoRA_{r=8}$.

## C  ADDITIONAL VISUALIZATION OF TASK EMBEDDINGS

Further analysis of task embeddings is presented in Figures 4-6. These figures reveal that the patterns observed in other layers and modules of the model are consistent with those reported in the main text. Notably, stronger clustering is observed in the $w_i$ and $w_o$ layers. This enhanced clustering may be attributed to the feed-forward network (FFN) layers' ability to capture shared information underlying different tasks more effectively.

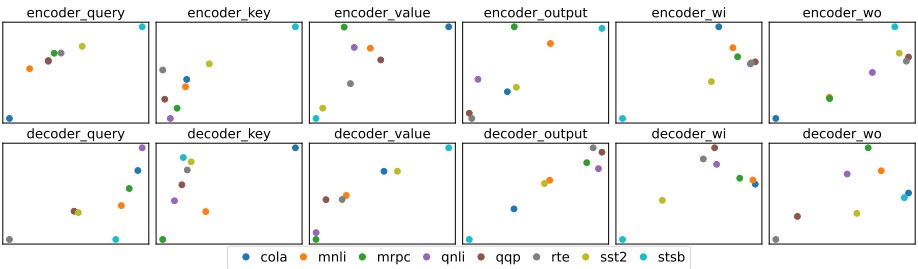

Figure 4: Visualization of Task Embeddings in Layer 1.

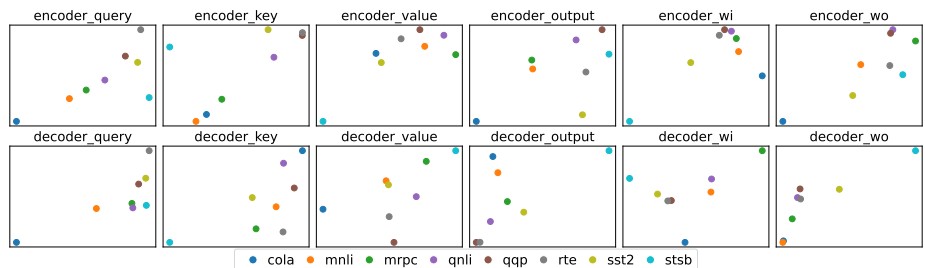

Figure 5: Visualization of Task Embeddings in Layer 6.

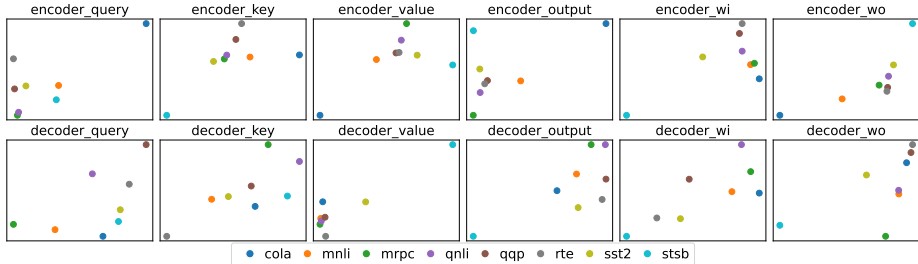

Figure 6: Visualization of Task Embeddings in Layer 12.

