# OpenReview forum: "MORE: A MIXTURE OF LOW-RANK EXPERTS FOR ADAPTIVE MULTI-TASK LEARNING"
_ICLR.cc/2025/Conference — Submitted to ICLR 2025_

### Official Review · Reviewer_6qqL · 2024-11-01

**Soundness:** 2
**Presentation:** 2
**Contribution:** 2
**Rating:** 3
**Confidence:** 5

**Summary:**

This paper introduces an approach to enhance the effectiveness of LoRA in multi-task Parameter Efficient Fine-Tuning scenarios. Traditional LoRA methods, which often employ a separate module for each task or focus solely on single-task applications, fall short in multi-task environments due to limited efficiency. MoRE addresses this by integrating multiple low-rank experts, each aligned with different tasks, within a single framework. More incorporates the adaptive rank selector, which dynamically chooses the most suitable expert for each specific task. This integrated approach allows for the simultaneous training of multiple low-rank experts.

**Strengths:**

1. The MoRE method introduces an approach by integrating multiple low-rank experts into a single, adaptive framework. This design, combined with an adaptive rank selector, provides a flexible and efficient solution for multi-task learning.

2. MoRE improves the overall performance of LLMs in multi-task settings by allowing multiple tasks to be addressed concurrently without the need for separate modules for each task.

**Weaknesses:**

1. One key advantage of LoRA is its ability to merge into the original model, which reduces inference costs. However, the use of a selector with varied ranks in this paper seems to compromise this advantage. Clarifying how this approach can be effectively merged back, if possible, or discussing strategies to mitigate the increased inference complexity would provide valuable insights and address a critical concern.

2. Testing the proposed method on more advanced models, such as Gemma2, would be beneficial to demonstrate its robustness and relevance.

3. The current experiments only test LoRA with a fixed rank of 8, which limits understanding of the proposed method’s performance under different rank configurations. Testing with a range of ranks would give a more comprehensive view of how varying rank values impact the approach's effectiveness and the trade-offs between parameter efficiency and task performance.

4. The method of setting the maximum rank for the selector is not clearly explained. It remains unclear whether the maximum rank is manually defined or set automatically. Providing a detailed explanation of this parameter choice and how it adapts to different tasks, if applicable, would enhance the method’s transparency and usability.

5. The performance improvement demonstrated by the proposed method appears marginal and does not achieve the best results across several tasks, including SST-2, MRPC, and ARC. Conducting statistical significance tests would help substantiate whether the reported improvements are meaningful and support the claims that the improvement is significant.

6. The relationship between the selected ranks, the specific tasks, and the reason are not well explored. A detailed analysis of how the selector determines ranks for each task and which type of tasks consistently require higher or lower ranks would provide valuable insights.

7. The current evaluation does not explore the effect of handling different numbers of tasks in a multi-task learning setting. Testing the model’s performance with varying numbers of concurrent tasks would provide valuable insights into its scalability and robustness.

**Questions:**

Please refer to the Weakness.

---

> ### Author Response · Authors · 2024-11-24
>
> Dear Reviewer 6qqL,
>
> **W1: Concern for the complexity of inference.**
> For methods based on MoE, integrating with the original model indeed presents some challenges, which is a common issue faced by all MoE-based approaches. The advantage of our method is that we have only one LoRA module, unlike other MoE methods that utilize multiple parallel LoRA modules. This design choice was initially made with consideration for inference speed. For fixed tasks, we can also choose to merge the experts corresponding to those tasks. In practical applications, merging LoRA is not necessarily required because the base model is typically fixed, but there may be many specialized LoRA modules that need to be switched on demand. Therefore, merging might not be the optimal solution. Our experts can also be viewed as multiple LoRA modules that can be switched as needed.
>
> **W2: Test performance on more advanced models.**
> We have conducted relevant experiments on the T5 model with encoder-decoder structure and the llama model with decoder-only structure, proving the effectiveness of our method. Based on your suggestions, we will also extend our testing to different large models for a more comprehensive experimental validation.
>
> **W3,W4: The setting of the maximum rank and testing only with rank 8.**
> Firstly, other LoRA methods have demonstrated that setting the rank to 8 often yields good results, so to ensure a fair comparison, we also set our rank to 8. Our experiments have confirmed that our method achieves better performance with the same amount of parameters. Furthermore, one of LoRA's advantages lies in its lower parameter count, making setting a commonly used rank both representative and practical. After setting the rank, the selection of rank experts for different tasks is achieved through training a routing module. By establishing task embeddings for each task, the model can calculate the rank expert corresponding to each task through the task embeddings, thereby achieving improved performance. Lastly, we also plan to experiment with different maximum ranks or explore more optimal rank selection methods to more comprehensively demonstrate the advantages of our approach.
>
> **W5: Improvements are not significant on some datasets.**
> When fine-tuning for multi-task scenarios, different tasks often have connections or conflicts. We address this by designing a contrastive learning module to learn the distinctions between tasks and using a routing module to adaptively assign rank experts to different tasks. This approach minimizes interference between tasks while allowing the same experts to better capture shared information across different tasks, thereby enhancing overall performance. In contrast, other methods might overfit on certain tasks, achieving good performance on individual datasets but falling short overall. This further demonstrates that our method can better balance performance across various datasets.
>
> **W6: The relationship between the selected ranks and the specific tasks.**
> We have set trainable task embeddings and used contrastive learning to constrain their training. Then, through a routing module, we select appropriate experts for different tasks. The routing module tends to choose the same experts for similar tasks. To better explore the rank selection of the model, we also display visualizations of the learned task feature embeddings and the rank distribution across different tasks in Figure 2. It can be observed that some tasks exhibit clear clustering. Regarding rank selection, in most model layers, the model tends to choose lower ranks for all tasks, suggesting a high degree of redundancy in the LoRA modules. However, for different tasks, it tends to select different higher ranks to showcase their uniqueness.
>
> **W7: The effect of handling different numbers of tasks in a multi-task learning setting.**
> We have established task embeddings for different tasks, which are scalable. More task embeddings can be set for additional tasks to participate in training. Additionally, our method can directly perform few-shot transfer for new tasks, as demonstrated by the experimental results in Table 4 of the paper.In practical applications, the number of multi-tasks is limited, allowing us to choose how to set task embeddings based on actual conditions to balance performance and speed.
>
> We look forward to your further feedback and hope that our responses meet your expectations.
>
> Best regards,
>
> The Authors

---

> > ### Comment · Reviewer_6qqL · 2024-11-28
> >
> > Thank you for your reply. While I appreciate the effort put into the response, it did not sufficiently address my concerns regarding the merge-back process, testing with more advanced models, the impact of LoRA rank, performance, and evaluation across various tasks. As these issues remain unresolved, I have decided to maintain my previous score.

---

### Official Review · Reviewer_ZeVE · 2024-11-03

**Soundness:** 2
**Presentation:** 3
**Contribution:** 2
**Rating:** 3
**Confidence:** 4

**Summary:**

This paper introduces the MoRA, a mixture of expert LoRA methods to leverage the MoE framework in parameter-efficient tuning settings. This paper is well-written and shows some gains on the performance. However, this paper lacks novelty compared with other related works and the method is only tested on GLUE benchmarks. Also, similar as other MoE LoRA methods, the lora experts cannot merge with the LLM during inference. Thus, this method will increase the memory usage during inference and cannot leverage inference optimization of the Transformer model.

**Strengths:**

This paper is well-written, and the statement is clear. This method improves previous LoRA MOE methods and provides some gains. The experiments show promising improvements compared to the baselines reported.

**Weaknesses:**

1. This paper only evaluates on GLUE benchmarks, where most of the datasets mainly focus on natural language processing instead of commonsense and complicated reasoning. The reasoning and knowledge capabilities of the LLMs are still under explored in this paper.

2. MoE-based LoRA uses more parameters than standard LoRA methods and this LoRA matrix cannot be merged into the original LLMs during inference. Therefore, during inference, the model requires more memory to perform inference and could not leverage inference optimization on Transformer model.

3. This approach has no novelty compared to other MoE lora methods.

**Questions:**

1. Whether does this method perform well on other popular reasoning or knowledge benchmarks such as MMLU, code or math?
2. How does this method perform inference?

---

> ### Author Response · Authors · 2024-11-24
>
> Dear Reviewer ZeVE:
>
> Thank you for your insightful comments and the time you have taken to review our manuscript. We highly appreciate your expertise and have addressed your concerns with detailed explanations below. We hope our responses clarify the aspects you highlighted and demonstrate the robustness and innovation of our approach.
>
>
> **W1,Q1: Lack of validation on commonsense and complicated reasoning datasets.**
> Existing LoRA-related methods often compare performance on the GLUE dataset and common sense reasoning datasets for fair comparison, and our approach is consistent with this. We have not only conducted experiments on the GLUE dataset but have also performed few-shot transfer experiments and experiments on common sense reasoning as shown in Table 3, achieving equally good results:
>  | Methods          | params/task | BoolQ | PIQA | OBQA | ARC-E | ARC-C | AVG  |
> |------------------|-------------|-------|------|------|-------|-------|------|
> | LLAMa2-LoRA      | 2.5M        | 80.9  | 77.7 | 79.0 | 83.7  | 76.9  | 79.6 |
> | LLAMa2-MultiLoRA | 10M         | 76.5  | 72.9 | 68.2 | 81.6  | 61.9  | 72.2 |
> | LLAMa2-MixLoRA   | 12.2M       | 84.3  | 79.5 | 82.6 | 86.8  | 76.3  | 81.9 |
> | LLAMa2-MOELoRA   | 4.5M        | 84.0  | 79.9 | 81.8 | 86.8  | 77.3  | 82.0 |
> | LLAMa2-MoRE      | 4.5M        | 87.2  | 82.3 | 83.0 | 86.7  | 74.2  | 82.7 |
>
> Additionally, datasets like MMLU, math, and code are typically used to test the performance of pre-trained models, whereas we focus on the performance of models after fine-tuning on downstream tasks. Therefore, we have not specifically trained on these types of datasets, as they are not our primary focus.
>
> **W2,W3: The novelty and the inference cost.**
> Under the same parameter count, MoE models have lower inference costs because not all parameters need to be activated during inference. However, most current MoE models often possess parameter volumes several times larger than the original LoRA, which is a point of concern for us. Instead of treating parallel LoRA modules as experts, we consider different rank spaces within a single LoRA module as distinct experts, achieving better performance than other MoE methods with fewer parameters. So compared to existing MoE methods, our innovation lies in not using additional parallel LoRA modules, thereby avoiding higher parameter costs. Furthermore, when the task is predetermined, we can selectively merge the corresponding experts into the original model, ensuring there is no inference delay. This capability is difficult to achieve with other MoE methods.
>
> **Q2: How to perform inference?**
> During inference, we only need to calculate the experts corresponding to the specific tasks. An expert is a subset of the A and B matrices in the LoRA module. After this, the inference process is no different from the original LoRA. Traditional MoE methods often require selecting the top-K LoRA module experts based on different input samples or tasks, which leads to higher inference costs.
>
> We look forward to your further feedback and hope that our responses meet your expectations.
>
>  Best regards,
>
>  The Authors

---

> > ### Comment · Reviewer_ZeVE · 2024-11-25
> >
> > Thanks for the response. However, most of my concerns are not addressed very well. I will keep the same score.

---

> > > ### Author Response · Authors · 2024-11-28
> > >
> > > Thank you for your reply. We are happy to engage in further discussion. Here, we will provide more details about the novelty of our method and the inference process.
> > >
> > > **Novelty:**
> > > Compared to LoRA, which uses a fixed rank to process all tasks, our approach offers greater flexibility. LoRA’s fixed-rank approach lacks adaptability, while we use a routing mechanism to select the appropriate expert for each task. This means that different tasks can be handled by experts with different ranks, which is our first advantage.
> > >
> > > Secondly, in comparison to other MoE methods, they typically use multiple parallel LoRA modules to process tasks. This results in more training parameters, increasing both training and inference costs. In contrast, our method does not require the introduction of multiple LoRA modules. We simply treat one LoRA module as multiple experts, significantly reducing the number of parameters. This is our second advantage.
> > >
> > > Finally, to better capture the features of each task, we have also introduced a contrastive learning module, which further improves our ability to learn task-specific representations.
> > >
> > > **Inference Process:**
> > > In the inference process, the expert selected for a specific task is determined because the task embeddings have already been trained. Once the task embedding is computed, the routing module will select the appropriate expert, and this expert is then used to perform inference for the current task. Therefore, once the specific expert is chosen, the inference process is essentially the same as the original LoRA.
> > >
> > > Compared to traditional MoE models, we do not need to select a tok-k module; instead, we simply choose a specific expert, and there is no parallel structure involved. As a result, the inference cost is not higher than that of the original LoRA.
> > >
> > > We hope this clarifies the advantages of our method and the efficiency of our inference process. If you have any further questions, we are happy to continue the discussion.

---

### Official Review · Reviewer_Rgja · 2024-11-04

**Soundness:** 3
**Presentation:** 2
**Contribution:** 3
**Rating:** 5
**Confidence:** 4

**Summary:**

In this paper, the authors aim to address the problem of Applying LoRA to multi-taslk scenarios. The authors propose a MoRE approach for multi-task PEFT, aligning different ranks of LoRA modules with different tasks using a Gate function and designing an adaptive rank selector to select the appropriate rank for each task. The empirical findings show that the proposed MoRE method can outperform existing multi-task LoRA methods.

**Strengths:**

(1)The topic of multi-task PEFT is important.

(2)Even though there are many typos appear in the current version, the paper is easy to follow.

(3) To use a gate function to define the selection of appropriate ranks for different tasks is interesting.

(4) The experiments are convinced

**Weaknesses:**

(1) The author should double-check their writings. For example: in Line 161, The target is to learn a shared model F with parameters θ to satisfy the requirements of different （？） simultaneously. And in Line 243 “One step further, during the backward pass, the arg max in Eq.(3) is non-differentiable” obviously, argmax appears in eq(4).

(2) The authors claim that a smaller learning rate will benefit the training of MoRE, it would be better to test different learning rate in the extensive experiments to support this claim.

(3) It would be better to test other LLMs, like Mistral-7B, to demonstrate the proposed method can be used in different architectures.

**Questions:**

Please see weaknesses.

---

> ### Author Response · Authors · 2024-11-24
>
> Dear Reviewer Rgja,
>
> Thank you for your thorough review and valuable feedback on our manuscript. We have taken your comments into serious consideration and have made several revisions to address the concerns raised. Please find below a detailed response to each of the issues you have mentioned.
>
>
> **W1: Spelling or descriptive errors.**
> Thank you for pointing out the spelling and descriptive inaccuracies in our manuscript. We have conducted a thorough review and meticulously revised the text to ensure clarity and correctness throughout. We have corrected all typographical errors and clarified any ambiguous descriptions to improve the manuscript's overall quality and readability.
>
>
> **W2: The benefit of a smaller learning rate needs to be verified.**
> In equation 7, we introduced a linearly scaled learning rate because the lower-layer experts are updated more frequently, thus requiring a smaller learning rate. We also verified the effectiveness of this setting through the ablation experiments conducted later. It can be seen as below (the row of w/o Linear Scaling):
>
>
> | Conditions            | GLUE Avg. |
> |-----------------------|-----------|
> | MoRE                  | 87.3      |
> | w/o Linear Scaling    | 87.0      |
> | w/o Task Embeddings   | 86.1      |
> | w/o CL optimization   | 86.3      |
> | w/o STE               | 86.4      |
> | w/ Random Sample      | 86.2      |
>
>
> **W3: The effectiveness on other models.**
> We have conducted relevant experiments on the T5 model with encoder-decoder structure and the llama model with decoder-only structure, proving the effectiveness of our method. Based on your suggestions, we will also extend our testing to different large models for a more comprehensive experimental validation.
>
> We look forward to your further feedback and hope that our responses meet your expectations.
>
> Best regards,
>
> The Authors

---

> > ### Comment · Reviewer_Rgja · 2024-11-27
> >
> > Thank you for your response, I acknowledge that it helps address some of my concerns.
> >
> > I have also checked other reviewers' comments and your corresponding responses. After careful consideration, I decide to lower my rating.
> >
> > Best regards.

---

> > > ### Author Response · Authors · 2024-11-28
> > >
> > > Many thanks for your response. We look forward to your detailed questions and concerns. And we will try our best to answer them.

---

### Official Review · Reviewer_8814 · 2024-11-04

**Soundness:** 2
**Presentation:** 2
**Contribution:** 2
**Rating:** 5
**Confidence:** 2

**Summary:**

This paper proposes a MoE LoRA-based method, MoRE, for parameter-efficient fine-tuning in multi-task scenarios. The authors introduce a shared low-rank matrix with different rank cut-offs, which serve as distinct experts. Additionally, they present an adaptive gating mechanism that selects the appropriate rank value for each task, enabling efficient specialization across diverse tasks.

**Strengths:**

1. The idea is interesting. Rather than allocating dedicated experts through separate low-rank adapters, this work reuses a single low-rank matrix, employing a gating function to select portions of it as distinct experts. This approach effectively models diverse tasks in multi-task training.

2. The experimental results are promising. The proposed method consistently outperforms baselines across most datasets, with particularly strong performance in the LLAMA setting.

**Weaknesses:**

1. The contribution over existing mixture-of-LoRA methods (such as MixLoRA and MOELoRA) appears limited. This work can be interpreted as a specific case within existing frameworks, where each expert has a rank of 1, and a condition is enforced such that, when selecting expert k, all preceding experts (1 to k-1) are also selected.

2. While the proposed method reuses the adapter matrix and claims that this reduces the number of parameters, making it independent of the number of experts, several concerns arise:
    - With this design, the first few rows or columns of the adapter matrix are reused by multiple tasks, which may be effective for tasks with some correlation, such as those in GLUE. However, this could be problematic when the feature-label mappings differ significantly across tasks.
    - The number of experts is constrained by the maximum rank of the LoRA matrix, which may lead to issues when the number of tasks is large.

**Questions:**

Q1: How does the proposed method handle scenarios where the number of tasks significantly exceeds the maximum rank of the shared low-rank matrix?

Q2: If tasks are highly distinct with no shared information, does the design fail to accommodate such diversity effectively?

Q3: Why is the selection enforced as 1:k rather than allowing an arbitrary subset of ranks from 1 to \( r_{\text{max}} \), which might seem more intuitive?

---

> ### Author Response · Authors · 2024-11-24
>
> Dear Reviewers 8814,
>
>
> Thank you very much for your valuable feedback and the opportunity to refine our manuscript. We appreciate the insights provided and have addressed each point in the following sections, where we elaborate on our contributions, discuss the impact of task similarity and the number of tasks, and justify our choices regarding the model architecture.
>
>
> **W1: Contribution and Novelty.** Our experts have different ranks, ranging from 1 to k. We do not have multiple rank-1 experts; instead, only one expert of a given rank is selected each time. The difference from existing MoE methods is that we do not use multiple parallel LoRA modules. Instead, we use subsets within a single LoRA module as experts, which greatly reduces training and inference costs.
>
>
> **W2,Q1,Q2: The impact of task similarity and excessive number of tasks.**
> Firstly, we treat a subset of the LoRA module as experts and define a trainable task embedding for each task. Then, through a routing module, we automatically select the appropriate experts for each task.Secondly, LoRA itself leverages the low-rank nature of large models, making it well-suited for multi-task scenarios. However, the original LoRA assigns a fixed rank to each task, which lacks flexibility. Our method can allocate more suitable ranks for different tasks, ensuring that the number of tasks does not impact performance.Finally, our approach comprehensively considers the relationships and differences between various tasks and employs contrastive learning to learn the features of different tasks, thereby selecting more appropriate experts to handle them.
>
>
> **Q3: Why not choose subset experts?**
> We have tried selecting subsets as experts, which can indeed significantly increase the number of the experts. However, experimental results were not very satisfactory. We suspect the possible reasons might be, on one hand, that choosing expert subsets makes the training of routing more challenging. On the other hand, selecting subsets can make it harder for tasks to share information. Allowing different experts to share some low-rank parameters aligns better with the structural design of neural networks. Lower layers more readily preserve shared general features, while higher layers are better suited for learning task-specific features. The experimental results are as follows:
>
> | Method/dataset     | MNLI | QQP  | QNLI | SST-2 | STS-B | MRPC | RTE  | COLA | AVG  |
> |--------------------|------|------|------|-------|-------|------|------|------|------|
> | T5-subset_experts  | 85.8 | 90.2 | 93.3 | 94.2  | 89.7  | 90.7 | 79.9 | 66.1 | 86.2 |
>
> We look forward to your further feedback and hope that our responses meet your expectations.
>
> Best regards,
>
> The Authors

---

> > ### Comment · Reviewer_8814 · 2024-11-27
> >
> > I appreciate the dedicated work that the authors invested in the rebuttal. Some of my concerns have been addressed, and I will keep my current rating for now.
> >
> > In particular, the authors addressed my concern regarding the subset of experts (Q3).
> >
> > However, for Q1, I previously mentioned:
> >
> > > This work can be interpreted as a specific case within existing frameworks, where each expert has a rank of 1, and a condition is enforced such that, when selecting expert \(k\), all preceding experts (1 to \(k-1\)) are also selected.
> >
> > The authors did not respond to this directly.
> >
> > For Q2, I asked how the proposed method can handle heterogeneous tasks and/or an excessive number of tasks.
> >
> > > Q2: If tasks are highly distinct with no shared information, does the design fail to accommodate such diversity effectively?
> >
> > However, the authors' response primarily discussed how the proposed method differs from LoRA, leaving my question unresolved:

---

> > > ### Author Response · Authors · 2024-11-28
> > >
> > > I apologize for the misunderstanding, and I would like to provide some clarifications.
> > >
> > > First, regarding W1, I apologize for the earlier misinterpretation of the question. Here, I would like to state the differences between our approach and the special case you mentioned. The biggest issue with other MoE (Mixture of Experts) methods is their high computational cost. Even if the rank of each expert is 1, it still requires individual computation for each expert to obtain the results, whereas our method only requires a single computation, which greatly reduces the computational cost. Additionally, because our experts share information in the low-rank part, we can more efficiently learn shared information across tasks. In contrast, multiple rank-1 experts do not have this advantage. Finally, with multiple rank-1 experts, the maximum rank they can handle during inference is 1. In contrast, our method can flexibly handle ranks from 1 to k, significantly enhancing the modeling capability of our approach and better meeting the requirements of the dataset.
> > >
> > > Regarding Q2, I apologize that our response did not meet your expectations. We are happy to engage in a detailed discussion. Originally, we intended to highlight the advantages of our method by comparing it with the original LoRA, as LoRA itself is already capable of being trained on multiple different tasks. However, we wanted to emphasize the improvements we made compared to LoRA in order to highlight our focus on this aspect. We have introduced task embeddings for different tasks, and we use a routing module to adaptively select experts, along with contrastive learning to constrain the learning of task embeddings. This allows us to better capture the representations of different tasks. Additionally, natural language itself contains some underlying consistency, such as syntactic structures, which means that the shared parts of our experts can also capture the shared information of these underlying logical structures. In cases where the task differences are too large, consider a special scenario where all tasks select expert k. In this case, our approach would align with LoRA with rank k, and our performance would not be worse than LoRA’s. However, our model is still capable of capturing some shared information, thus achieving better performance than LoRA.

---

### Meta-Review · Area_Chair_KwT5 · 2024-12-13

**Metareview:**

In this paper, the authors proposed a new parameter fine-tuning algorithm for multi-task learning.

There are a number of major concerns raised by the reviewers: 1, Technically, the proposed algorithm is incremental, which combines existing techniques for multi-task parameter fine-tuning; 2, Some design of the proposed algorithm is not questionable; 3, The experimental results are not convincing.

The authors failed to address the concerns during the rebuttal. Therefore, based on its current shape, this paper is not ready for publication.

**Additional Comments On Reviewer Discussion:**

The concerns raised by the reviewers are not well-addressed during the rebuttal.

---

### Decision · Program_Chairs · 2025-01-22

Reject